# Three years of sea ice freeboard, snow depth, and ice thickness of the Weddell Sea from Operation IceBridge and CryoSat-2

Ron Kwok[1], Sahra Kacimi[1]

[1]Jet Propulsion Laboratory, California Institute of Technology

*Correspondence to*: Ron Kwok (ron.kwok@jpl.nasa.gov)

**Abstract.** We examine the variability of sea ice freeboard, snow depth, and ice thickness in three years (2011, 2014, and 2016) of repeat surveys of an IceBridge (OIB) transect across the Weddell Sea. Averaged over this transect, ice thickness ranges from 2.40±1.07 (2011) to 2.60±1.15 m (2014), and snow depth from 35.8±11.5 (2016) to 43.6±10.2 cm (2014); suggesting a highly variable but broadly thicker ice cover compared to that inferred from drilling and ship-based
measurements. Spatially, snow depth and ice thickness are higher in the more deformed ice of the western Weddell. The impact of under-sampling the thin end of the snow depth distribution on these spatial statistics, due to the resolution of the snow radar, is assessed. Radar freeboards (uncompensated for snow thickness) from CryoSat-2 (CS-2), sampled along the same transect, are consistently higher (by up to 8 cm) than those computed using OIB data. This suggests radar scattering that originates above the snow-ice interface, possibly due to salinity in the basal layer of the snow column. Consequently, sea
ice thickness computed using snow depth estimates solely from differencing OIB and CS-2 freeboards (without snow radar) are therefore generally higher; mean differences in sea ice thickness along a transect are up to ~0.6 m higher (in 2014). This analysis is relevant to the use of differences between ICESat-2 and CS-2 freeboards to estimate snow depth for ice thickness calculations. Our analysis also suggests that, even with these expected biases, this is an improvement over the assumption that snow depth is equal to the total freeboard, where the underestimation of thickness could be up to a meter. Importantly,
better characterization of the source of these biases is critical for obtaining improved estimates and understanding limits of retrievals of Weddell Sea ice thickness from satellite altimeters.


# 1    Introduction

As snow loading is required for conversion of freeboard to thickness, the reliable determination of sea ice thickness in the Antarctic remains a challenge largely due to uncertainties in snow depth (e.g., Giles et al., 2008). In the Antarctic, as in the Arctic, such estimates of sea ice thickness are necessary to evaluate both stand-alone sea ice and coupled climate models, attribute the causes of recent observed changes, evaluate and understand the physical processes controlling sea ice extent and thickness, and to improve model projections of the future sea ice cover. One distinguishing feature of sea ice in the Antarctic vis-à-vis the Arctic is the common occurrence of snow-ice due to heavier snow fall over Antarctic sea ice (Massom et al., 2001): when snow load depresses the ice surface of the thinner Antarctic sea ice below sea level, seawater infiltrating the base of the snow layer leads to the formation snow-ice when the resultant slush freezes. The changes in the properties of the snow layer due to the thicker snow cover, flooding, and snow-to-ice conversion, complicate the large-scale retrievals of snow depth and ice thickness. In recent efforts to estimate Antarctic sea ice thickness using various estimates of snow depth, a wide discrepancy between ice thickness estimates persists (Yi et al., 2011; Kurtz and Markus, 2012; Xie et al., 2013).

Since 2009, NASA's Operation IceBridge (OIB) (Koenig et al., 2010) has flown surveys to acquire spring data over the Arctic and Antarctic sea ice covers. Implemented as an airborne remote sensing program to extend the laser altimeter time-series through the gap between the end of the ICESat mission and the launch of the ICESat-2 (IS-2) lidar, OIB has acquired a unique time series that allows for examination of the interannual behavior of Antarctic sea ice cover as well as a better understanding of the remote sensing issues associated with the retrieval of sea ice freeboard and thickness. In addition to a lidar to determine freeboard, the OIB instrument suite includes an ultra-wideband radar that is capable of resolving the location of the air-snow and snow-ice interfaces, and hence providing snow depth estimates. Over the OIB mission, the sensitivity of snow depth retrievals to associated snow properties (density and salinity) has become a special emphasis because of the significant impact of snow on thickness estimates using lidar or radar (Kwok, 2014). Except for a recent analysis that specifically addressed the variability of OIB snow depths in the Weddell and Bellingshausen Seas (Kwok and Maksym, 2014), this OIB data set has received less attention relative to the data acquired over Arctic sea ice. In this paper, we examine the spatial and interannual variability of snow depth and sea ice thickness from the repeat survey of an OIB track in the Weddell Sea and use the combined OIB and CryoSat-2 (CS-2) data to inform the derivation of snow depth and ice thickness from satellite altimetry.

With the planned launch of IS-2 (Markus et al., 2016) (scheduled for late 2018) to continue the altimetry time series to inform changes in the cryosphere, there may be a unique opportunity to obtain near-coincident altimetry of the sea ice cover from both a lidar (IS-2) and a radar (CS-2), especially for the extraction of snow depth for thickness calculations. In an analysis using OIB and CS-2 acquisitions over the Arctic Ocean, Kwok and Markus (2017) demonstrated that snow depth can potentially be estimated from differencing the lidar and radar freeboards. Hence, it would be of particular interest if such an analysis approach could be used in the Antarctic as well.

In this paper, we address two topics: 1) the spatial and interannual variability of total freeboard, snow depth, and sea ice thickness in three years (2011, 2014, and 2016) of repeated IceBridge surveys of a transect across the Weddell Sea, and

2) the estimation of sea ice thickness, in the absence of snow depth measurements, using only freeboards from the IS-2 and near-coincident CS-2 radar freeboards. The paper is organized as follows. Section 2 describes the instruments and the data sets used in our analyses. Section 3 outlines the different ways of calculating freeboard, snow depth, and thickness using estimates from lidar and radar freeboards. Section 4 summarizes the spatial and interannual variability of these three sea ice

parameters in the three years of OIB data. Section 5 discusses estimates of ice thickness, in the absence of snow depth measurements, using only estimates of freeboard from the ATM lidar and near-coincident CS-2 radar freeboards. The quality of thickness estimates is assessed by comparison with those estimates calculated with the assumption of snow depth is equal to the total freeboard (i.e., zero ice freeboard). Section 6 summarizes the results that are of geophysical interest.

## 2    Data Description

The three data sets of interest are the total (snow+ice) freeboard from the Airborne Topographic Mapper (ATM), the snow depth estimates from the ultra-wideband snow radar (SR) from OIB, and the radar freeboard from CS-2. All through the OIB campaigns, the ATM and SR instruments have been operated simultaneously and provide near coincident coverage, albeit at different spatial resolutions. In this section, we provide a brief description of the specifications and coverage of these lidar and radar systems, and the quality of the retrievals.

### 2.1    IceBridge ATM freeboard

Surface elevation are from the IceBridge Narrow and Wide Swath ATM Level-1B lidar Elevation and Return Strength data set (Krabill, 2014). This data set contains ATM spot elevation measurements ($\sim$ 1-2 m footprint) over sea ice. ATM scanning geometry provides an across-track scan swath of 45/250 m with typical elevation accuracy for individual samples to be better than 10 cm. Total freeboard (ice+snow, $h_f$), the height of the snow surface above the local sea surface,

from the lidar were derived from the IceBridge elevation data using the approach described by (Kwok et al., 2012). Sea surface references are identified in high-resolution visible imagery acquired by the Digital Mapping System (Dominguez, 2010). For the analysis here, total freeboard of an elevation sample is calculated only when open water surfaces (i.e., leads) within 10 km are present to serve as local sea level reference. We use at least 10 samples (but the population is typically higher) to estimate each sea surface reference, giving a precision of typically better than 3 cm.

### 2.2    IceBridge snow depth

Estimates of snow depth were derived from the IceBridge Level-1B Radar Echo Strength Profiles data set (Leuschen, 2014) from the snow radar. This frequency-modulated continuous-wave (FM-CW) radar is operated by the Center for Remote Sensing of Ice Sheets (CReSIS) at the University of Kansas. The large bandwidth ($\sim$6 GHz) provides a range resolution of $\sim$5 cm (in free space) for resolving the location of the air-snow (a-s) and snow-ice (s-i) interfaces (Panzer et al.,

2013). With averaging, the spot separation is $\sim$1 m along track at an altitude of $\sim$500 m and an air speed of $\sim$250 kts (the nominal flight parameters for all OIB sea ice surveys). The size of the average footprint is $\sim$5–10 m, and the spacing between

the processed radar profiles is ~5 m. The reader is referred to the published literature for a more detailed description of the radar system (e.g., Panzer et al., 2013) and of the data characteristics (e.g., Kwok et al., 2011).

Snow depth is calculated using a simplified version retrieval procedure described by Kwok and Maksym (2014) that has the capability to compensate for effects due to residual system sidelobes in the returns (Kwok and Haas, 2015). A bulk snow density of 320 kg/m$^3$ was used to convert the range differences between the air-snow (a-s) and snow-ice (s-i) interfaces (in free space) to snow depth. Since a re-processed version of the radar data set with suppressed system sidelobes is now available for all years, the aspect of the algorithm that deals with system sidelobes has been disabled. In this algorithm (described in Kwok and Maksym, 2014), both the snow-ice and air-snow interfaces are detected and localized by determining the significance of each local peak above the noise floor in individual echo returns. Significance is determined by the strength and width of the local maxima (power) and its associated leading/trailing edges relative to the expected noise power of the system. The system bandwidth controls the width (or sharpness) of a local maximum and the rate of rise of its leading edge. The algorithm uses these system-dependent parameters to adapt to the changes in the radar system as the bandwidth and noise level of the snow radar have progressively improved over the course of the OIB mission. The highest significant peak in the echo profile is designated as the return from the s-i interface. Returns from a-s interfaces are assumed to be weaker and the first significant range returns, determined using the above criteria, above the s-i interface. Once the interfaces are detected, the radar range to the interface is localized in an oversampled (by 16 times) version of the echo return; this reduces the range error in the identification of the local maxima in the echo return. From a scattering perspective, this restricts the detected returns to a-s interfaces that are more specular and appear as a detectable peak, rather than just the strength of the leading edge.

Assessment of the snow depth retrievals in the Arctic with in-situ measurements from two field programs shows that they are within −1.8±3.4 and -2.2±4.6 cm of those obtained with magna-probes (see Kwok et al., 2017). Additionally, the bandwidth of the radar system (or range resolution) imposes a lower bound on the resolvable snow depth to ~8 cm.

## 2.3 CryoSat-2 freeboard

Along-track CS-2 freeboards are those from Kwok and Cunningham (2015). The reader is referred to Kwok and Cunningham (2015) for a more detailed description of the retrieval procedures and quality of these fields. As there are no direct freeboard estimates, comparisons to available ice thickness measurements provide an indirect measure of quality: freeboard is approximately ~one-ninth of ice thickness. The assessed differences between CS-2 and various thickness measurements are: 0.06±0.29 m (ice draft from moorings), 0.07±0.44 m (submarine ice draft), 0.12±0.82 m (airborne electromagnetic profiles), and -0.16±0.87 m (Operation IceBridge).

## 3 Derived estimates of freeboard, snow depth, and thickness

In this section, we outline different ways by which one can derive freeboard, snow depth, and thickness from the three retrieved quantities described in Section 2 (two from OIB and one from CS-2). The three quantities are: 1) the total freeboard

(i.e., snow+ice) from the ATM lidar ($h_f^{ATM}$), 2) snow depth from the snow radar ($h_{fs}^{SR}$), and 3) the radar freeboard ($h_{fi}^{CS2}$) from CS-2. Here, we define derived versus retrieved quantities: derived estimates, referred to in the balance of this paper and this section, are quantities calculated from the retrieved quantities. The significance of these derived estimates is discussed. In Section 5, we examine the differences between the retrieved and derived freeboards, snow depths, and their relative

5    impact on thickness calculations.

### 3.1    Snow depth from differences in lidar and radar freeboards

For a simple layered system in Figure 1, snow depth can be estimated as the difference between the retrieved ATM ($h_f^{ATM}$) and CS-2 ($h_{fi}^{CS2}$) freeboards (Kwok and Markus, 2017),

$$\tilde{h}_{fs} = \frac{(h_f^{ATM} - h_{fi}^{CS2})}{\eta_s} \tag{1}$$

10   where $\eta_s = c/c_s(\rho_s)$, and $c/c_s(\rho_s) = (1 + 0.51\rho_s)^{1.5}$. The superscript 'tilde' denotes a derived estimate calculated from retrieved quantities (as defined above). The adjustment (scaling by $\frac{1}{\eta_s}$) compensates for the reduced propagation speed of the radar wave ($c_s$) in a snow layer with bulk density $\rho_s$.

Comparison of the derived ($\tilde{h}_{fs}$) with the retrieved snow depth ($h_{fs}^{SR}$) tells us how well we can estimate snow depths using the differences between lidar ($h_f^{ATM}$) and radar freeboards ($h_{fi}^{CS2}$) in the absence of a snow-radar. This is of particular

15   interest since near coincident observations of freeboards from CS-2 (a radar altimeter) and IS-2 are potentially useful for providing large-scale estimates of snow depth (Kwok and Markus, 2017).

### 3.2    Total and ice freeboards

The total freeboard ($\tilde{h}_f$) can be derived by combining the retrieved CS-2 freeboard ($h_{fi}^{CS2}$) with retrieved snow depth ($h_{fs}^{SR}$), or with an estimate of the snow depth from differencing the lidar and radar freeboard ($\tilde{h}_{fs}$ in Equation 1):

$$\tilde{h}_f = h_{fi}^{CS2} + \eta_s h_{fs}^{SR} \tag{2}$$

$$\tilde{h}_f' = h_{fi}^{CS2} + \eta_s \tilde{h}_{fs} \tag{3}$$

The prime next to a variable (e.g., $\tilde{h}'$) indicates that the estimate, $\tilde{h}$, is based on snow depth computed by differencing ATM and CS-2 freeboards (i.e., $\tilde{h}_{fs}$), instead of the retrieved value from the snow radar; this allows us to identify the

quantities that are calculated using a combination of derived and retrieved quantities, rather than just retrieved quantities. By comparing these derived freeboards with $h_f^{ATM}$, the quality of the derived estimate can be assessed.

Similarly, the ice freeboards ($h_{fi}$, i.e., the height of the snow-ice interface above the local sea surface) can be derived as follows,

$$\tilde{h}_{fi} = h_{fi}^{CS2} + h_{fs}^{SR}(\eta_s - 1) \tag{4}$$

$$\tilde{h}'_{fi} = h_{fi}^{CS2} + \tilde{h}_{fs}(\eta_s - 1). \tag{5}$$

As above, a primed variable (e.g., $\tilde{h}'$) is one that is based on snow depth computed by differencing ATM and CS-2 freeboards (i.e., $\tilde{h}_{fs}$).

The expected CS-2 or radar freeboard ($\tilde{h}_{fi}^{CS2}$) is calculated from OIB snow depth ($h_{fs}^{SR}$) and total freeboard ($h_f^{ATM}$) as:

$$\tilde{h}_{fi}^{CS2} = h_f^{ATM} - \eta_s h_{fs}^{SR}. \tag{6}$$

This is of interest because the comparison of derived radar freeboard ($\tilde{h}_{fi}^{CS2}$), from the OIB data, with the retrieved radar freeboard ($h_{fi}^{CS2}$) may provide insights into the penetration of the radar wave into the snow layer. If the radar returns originate above the snow-ice interface due to brine or layering in the snow volume (discussed later), this will have the effect of increasing the retrieved radar freeboard (i.e., $h_{fi}^{CS2} > \tilde{h}_{fi}^{CS2}$) or lowering the derived snow depth ($\tilde{h}_{fs}$). The consequence is an overestimation of total and ice freeboards ($\tilde{h}_f, \tilde{h}'_f, \tilde{h}_{fi}, \tilde{h}'_{fi}$ in the above equations) and therefore the sea ice thickness.

### 3.3    Sea ice thickness

Assuming hydrostatic equilibrium, sea ice thickness ($h_i$) from total ($h_f$) or ice freeboards ($h_{fi}$), with a snow layer of thickness $h_{fs}$, can be calculated as follows (see geometry in Figure 1):

$$h_i^{lidar}(h_f, h_{fs}) = \left(\frac{\rho_w}{\rho_w - \rho_i}\right)h_f - \left(\frac{\rho_s + \rho_w}{\rho_w - \rho_i}\right)h_{fs} \quad \text{(lidar)} \tag{7}$$

$$h_i^{radar}(h_{fi}, h_{fs}) = \left(\frac{\rho_w}{\rho_w - \rho_i}\right)h_f + \left(\frac{\rho_s}{\rho_w - \rho_i}\right)h_{fs} \quad \text{(radar)} \tag{8}$$

$\rho_s$, $\rho_w$, and $\rho_i$ are the bulk densities of snow, water and ice, respectively. These equations are written slightly differently to show their explicit dependence on the lidar and radar observables – $h_f$ and $h_{fi}$.

With the retrieved and derived estimates of freeboard and snow depth from above, sea ice thickness can be calculated in six different ways using a combination of retrieved and derived quantities: $h_i^{lidar}(h_f^{ATM}, h_{fs}^{SR})$, $h_i^{lidar}(h_f^{ATM}, \tilde{h}_{fs})$, $h_i^{radar}(\tilde{h}_{fi}, h_{fs}^{SR})$, $h_i^{radar}(\tilde{h}_{fi}, \tilde{h}_{fs})$, $h_i^{radar}(\tilde{h}'_{fi}, h_{fs}^{SR})$, and $h_i^{radar}(\tilde{h}'_{fi}, \tilde{h}_{fs})$. For the data set considered here, the best estimates of thickness are those using only OIB measurements, i.e., $h_i^{lidar}(h_f^{ATM}, h_{fs}^{SR})$. The third through the fifth thickness estimates (i.e., $h_i^{radar}(\tilde{h}_{fi}, h_{fs}^{SR})$,

$h_i^{radar}(\tilde{h}_{fi}, \tilde{h}_{fs})$, and $h_i^{radar}(\tilde{h}'_{fi}, h_{fs}^{SR})$) depend on all three retrieved variables, while the other three depend only on two. If snow depths from the snow-radar were not available, then the sea ice thickness can be estimated using $h_i^{lidar}(h_f^{ATM}, \tilde{h}_{fs}) = h_i^{radar}(\tilde{h}'_{fi}, \tilde{h}_{fs})$. As mentioned above, this is of particular interest if we were able to obtain near coincident observations from a radar altimeter (e.g., CS-2) and a laser altimeter for derivation of $\tilde{h}_{fs}$ – a possibility after the launch of IS-2.

**4    Weddell Sea: Freeboard, snow depth and ice thickness from OIB**

In this section, we examine the repeat surveys of an OIB transect of the Weddell Sea ice cover: twice in 2011, and once in 2014 and 2016. All four flights were flown in October. These OIB acquisitions represent the first large-scale airborne surveys of freeboard and snow depth of the Weddell Sea ice cover. The ~3700 km flight track starts at a point just east of the northern tip of the Antarctic Peninsula (see Figure 2). From there, the eastbound leg (A to B) crosses the Weddell Sea

(~1500 km) to Cape Norvegia. The second leg (B to C) heads southwest hugging the coast before turning west, south of the Brunt Ice Shelf, for the westbound or return crossing (C to D). The track ends just south of James Ross Island in the western Weddell Sea. The scientific objective of this OIB loop was to sample the east-west gradient in Weddell Sea ice thickness in the outbound as well as return legs.

A detailed analysis of the consistency in snow depth distributions in the two 2011 repeat tracks can be found in Kwok

and Maksym (2014). Here, we first discuss the spatial and interannual variability of freeboard and snow depth from the ATM lidar and snow radar, and then the ice thickness derived from these retrieved quantities (Figure 2).

**4.1    Freeboard and snow depth**

Even though the variability is quite high, for all three years the snow depths and total freeboards (Table 1 and Figure 2) are generally higher in the western Weddell Sea (west of 45°W). As a reminder, snow depth is the retrieved parameter from

the snow radar ($h_{fs}^{SR}$) and total freeboard is from the ATM lidar ($h_f^{ATM}$). This ice cover then transitions into a region of thinner snow and freeboard in the eastern Weddell (at ~1300 km along the track). In the westbound legs, the large-scale trends are reversed, i.e., the snow and freeboard thickens as the flight track approaches the western Weddell and the coast of the Antarctic Peninsula. For the three years, this east-west gradient is seen in both the outbound legs (A to B) in the north as well as the return legs (C to D) farther to the south.

Averaged over the entire transect, the snow depths (in cm) are 36.3±11.7 (Oct 11, 2011), 35.2±11.0 (Oct 25, 2011), 43.6±10.2 (Oct 20, 2014), and 30.0±8.51 (Oct 27, 2016) (see Table 1), and the total freeboards (in cm) are 49.3±17.2 (Oct 11, 2011), 49.8±17.9 (Oct 25, 2011), 56.3±17.2 (Oct 20, 2014), and 45.4±16.3 cm (Oct 27, 2016). Although the data set covers only three years, it provides some indication of interannual variability. It is interesting to note the correspondence of

higher snow depths and freeboards in 2014, and lower snow depths and freeboards in 2016. In all but a few samples (2 to 3 12.5-km samples) total freeboards are higher than snow depths (on average ~13 cm), suggesting limited areas where the consequence of snow loading leads to zero or near-zero ice freeboard.

The 12.5-km averages of snow depth shown here range from ~50 cm near the Antarctic Peninsula to ~10 cm in the eastern Weddell. Consistent with that reported in Kwok and Maksym (2014) (where they reported extremes in 4 km averages), the snow depths in all three years are generally higher than the vast majority of those reported from in situ data

(Massom et al., 2001). As noted in Kwok and Maksym (2014), field observations of snow depth from two sources – underway shipboard observations and mechanical drilling profiles – favor sampling of the thinner end of the snow depth distribution due to physical and logistical constraints; thus, these sample populations may not be representative of regional statistics. Furthermore, the sea ice cover sampled by the OIB tracks has rarely been surveyed this late in the season (i.e.,

October), in part because of restricted ship accessibility to these areas with thicker ice and snow. By the time of these OIB surveys, the ice cover will have experienced the full season of growth, deformation, and snow accumulation. We attribute the differences seen here, in large part, to spatial sampling constraints inherent in available field measurements and to the lack of observations during spring in much of the Weddell Sea.

For all three years, the snow depth retrievals suggest that much of the region in the interior pack in spring, particularly

near the Antarctic Peninsula, has much deeper snow (and, thicker ice, see below) than has been typically described elsewhere. The regions of heavy deformation and multiyear ice in the western Weddell Sea have rarely been sampled (particularly in spring), and these ice types are well known to possess deep snow covers. Potentially, due to the resolution limitations of the snow radar (lower bound on the resolvable snow depth is ~8 cm), the areas of thin snow and ice are under-sampled by our retrieval process but this does not preclude the fact that areas of much thicker snow are seen in the data and

therefore sampling for obtaining regional statistics remains an issue.

## 4.2    Sea ice thickness

Averaged over this transect, ice thickness ranges between 2.40±1.07 and 2.51±1.16 (October, 2011) and 2.60±1.15 m (October 2014), and 2.44±1.18 m (October 2016) (Table 2 and Figure 2), strikingly similar for the three years. As with the east-west gradient in snow depth and freeboard, the thicker sea ice is in the more deformed ice cover in the western Weddell

Sea. Another point of note is that even though there is a 13 cm difference in the average snow depth between 2014 and 2016, the difference in thickness between the two years is only 0.16 m because the difference in total freeboard is only 11 cm, giving a small overall thickness change. The 12.5-km averages of ice thickness shown here range from ~5 m near the Antarctic Peninsula to ~1 m in the eastern Weddell.

Recent surveys of ice draft in the northwest Weddell Sea using autonomous underwater vehicles (Williams et al., 2015) report highly variable ice drafts in a deformed ice cover with a mean draft of 2.40±1.68 m and a maximum draft exceeding 14 m. The authors note that their measurements in the Weddell Sea are much higher than that reported by most drilling (mean/std draft: 1.05±0.4 m) and ship-based measurements (mean/std draft: 1.01±0.5 m) (see Table 1 in Williams et al., 2015). Our thickness averages also suggest, on an even broader spatial and temporal scale, a thicker and higher variable Weddell Sea ice cover than those inferred from drilling and ship-based measurements. Again, this highlights the potential limitations of shipboard observations and drilling as they favor sampling of the thinner end of snow and thickness distributions due to physical and logistical constraints.

### 4.3    Sampling of the snow depth distribution and ice thickness

As mentioned in Section 2, the bandwidth of the radar system (or range resolution) imposes a lower bound on the resolvable snow depth to ~8-10 cm. Further, over fairly rough surfaces, the retrievals tend to be discarded as unreliable (Kwok et al., 2011). These omissions at the two extremes cause an incomplete sampling of the snow depth distribution, and thus the question arises as to how these shortcomings in sampling affect the statistics of the snow depth distribution, and the calculated ice thickness when total freeboards are used. That is, how representative are the spatial averages discussed above. One way to address the potential of sampling biases was employed by Kwok and Maksym (2014): they used tabulated relationships between OIB snow depth and total freeboard (i.e., $h_{fs}^{SR} = f(h_f^{ATM})$) to estimate snow depths at lidar samples (i.e., total freeboard) where there were no retrievals. We follow this approach: we fill the samples with no snow depth retrievals and then we compare the snow depth distributions and the thickness estimates before and after filling the samples. The results are show in Figure 3.

For the repeat Weddell tracks, the averaged snow depths (after filling) are lower between 4.0 cm (in 2011) and 6.0 cm (in 2014), largely due to the undersampling of thin snow – a limitation due to the resolution of the snow radar. The resulting ice thickness estimates are actually higher (between 0.09 m in 2016 and 0.37 m in 2014). This is somewhat counter-intuitive, as reduced snow loading should imply lower ice thickness for a fixed ice freeboard. Here, however, we calculate thickness with averaged along-track total freeboard and there may be missing snow depth retrievals in the corresponding along-track snow depth averages. As the ice freeboard is the difference between total freeboard and snow depth, the ice freeboard becomes higher when the snow depth is lowered (due to more complete sampling of the thin end of snow depth distribution) hence increasing the ice thickness estimates.

In this case, the results from this simple assessment suggest that, in our current calculations, the spatially averaged snow depths may be overestimated while ice thickness may be underestimated if we did not account for the thin end of the snow depth distribution.

## 5    Estimates of snow depth and ice thickness using lidar and radar freeboards

In this section, we address the use of ATM and CS-2 freeboards to estimate snow depth and ice thickness with the aim of identifying potential biases in radar freeboard and snow depth that may impact thickness estimates. First, we discuss the construction of CS-2 freeboard estimates at the ATM ground tracks. Second, we compare the retrieved CS-2 radar freeboards with those derived from OIB retrievals (lidar and snow depth). Third, we compare the derived snow depths ( $\tilde{h}_{fs}$ ), total freeboard ( $\tilde{h}'_f$ ) and ice thickness, with those computed using the retrieved quantities from OIB.  Last, we examine the potential biases in thickness estimates if only lidar freeboards were available.

### 5.1    Co-locating CS-2 and ATM freeboards

Because of the disparity in spatial resolution and the space-time sampling of the Antarctic ice cover by the CS-2 altimeter, the OIB ATM lidar and the snow radar, the first step of the process is to construct space-time averages with sample populations that are large enough to support the analysis undertaken here. We follow the procedure in Kwok and Markus (2017), where  CS-2 freeboards, interpolated to the ATM track locations, are from 30-day gridded fields (12.5 km by 12.5 km) centered on the day of each of the OIB flights.

The choice of sampling for these comparisons is governed by limitations of the two data sets (OIB ATM and SR, and CS-2). In particular, the OIB lidar data are acquired only during October and the sampling of the OIB flightlines are generally not aligned with the CS-2 ground tracks in time and space. Since the CS-2 ground tracks do not provide dense coverage of the surface, we are dependent on comparing spatial averages in the monthly CS-2 freeboard composites with the along-track freeboard averages from the OIB mission (see Figure 4).

### 5.2    Comparison of CS-2 freeboard estimates

Derived radar CS-2 freeboards ( $\tilde{h}_{fi}^{CS2}$ ) along the transect are compared with the retrieved radar freeboards ( $h_{fi}^{CS2}$ ) in Figures 4 and 5. $\tilde{h}_{fi}^{CS2}$ is calculated with Equation 6 using total freeboards from the ATM lidar ( $h_f^{ATM}$ ) and snow depths from the snow radar ( $h_{fs}^{SR}$ ).  For the three years, mean $h_{fi}^{CS2}$ ranges from ~11 cm to 14 cm, while $\tilde{h}_{fi}^{CS2}$ ranges from ~5 to 9 cm; their standard deviations (6 to 9 cm) are comparable (Table 1). The CS-2 radar freeboards ( $h_{fi}^{CS2}$ ) are consistently higher than the derived radar freeboards ( $\tilde{h}_{fi}^{CS2}$ ). Differences along individual tracks are: 5.96±10.3 and 4.05±9.31 (two flights in Oct 2011), 8.45±7.08 (Oct 2014), 1.21±8.22 (Oct 2016) cm (see also Table 3).

Equation 6 assumes that the difference between $h_f^{ATM}$ and $h_{fi}^{CS2}$ can be explained entirely by the reduced propagation speed of the radar wave in a snow layer. If the radar returns are from above the snow-ice interface due to salinity or layering in the snow volume, this will have the effect of increasing the radar freeboard (i.e., $h_{fi}^{CS2} > \tilde{h}_{fi}^{CS2}$ ). While these differences

may be due to ATM freeboard or snow depth retrievals, the magnitude of these differences (except for 2016) seem too high to be attributable to $h_f^{ATM}$ or $h_{fs}^{SR}$ (as discussed in Section 2), suggesting that retrieved $h_{fi}^{CS2}$ may indeed be higher than the expected ice freeboard.

The agreement in the direction of the bias (i.e., $h_{fi}^{CS2} > \tilde{h}_{fi}^{CS2}$) over the three years may be fortuitous, but other evidence

also point to the displacement of the scattering surface away from the snow-ice interface. In an assessment of ERS-2 radar altimetry (Ku-band: also CS-2 frequency) over the Weddell Sea, Giles et al. (2008) also found that the radar freeboards from ERS are higher than expected ice freeboards, which suggested that the radar may not be penetrating to the snow-ice interface. Field studies of K$_u$-band (CS-2 frequency) penetration into the snow cover (Willatt et al., 2010) also reported that the snow-ice interface was the dominant scattering surface only for snow without morphological features or flooding. In a

review article of snow on Antarctic sea ice, Massom et al. (2001) report that as a result of capillary suction of brine and flooding, high salinities (> 10 psu) occur up to about 0.1 m in the snow column, but mainly in the basal layer 0-5 cm layer above the ice surface. A recent analysis by Nandan et al. (2017) indicates that saline snow above the snow-ice interface on Arctic sea ice (observed on fast ice in the Canadian Arctic Archipelago) may indeed mask the contribution of scattering of the snow-ice interface to the radar return by effectively reducing  the penetration into the snow volume, hence affecting

thickness estimates using radar freeboards. While the processes associated with saline snow may be different in the Southern Ocean ice cover (perhaps saline snow associated with flooding as noted by Willatt et al. (2010) and (Massom et al., 2001)), the results suggest a source of bias in the radar returns worthy of further investigation.

The consequence is an overestimation of total and ice freeboards ($\tilde{h}_f$, $\tilde{h}_f'$, $\tilde{h}_{fi}$, $\tilde{h}_{fi}'$ in equations above) and therefore the sea ice thickness using Equations 7 and 8. Over the three years, these results also suggest that the biases may be dependent

on snow loading as the biases are higher in 2014 where the mean snow depth was highest (mean: 43.6 cm) and the biases are lower in 2016 where the mean snow depth was lowest (mean: ~30.0 cm). But, a more extensive data set would be required to quantify and substantiate this dependence.

### 5.3      Comparison of snow depth, freeboard, and thickness estimates

When the retrieved $h_{fi}^{CS2}$ are higher than expected (see above), the derived snow depths ($\tilde{h}_{fs}$) calculated using Equation

1 are lower, and the derived total freeboards ($\tilde{h}_f$) calculated using Equation 2 are higher (Figures 4 and 5, Table 3). The differences between $\tilde{h}_{fs}$ and $h_{fs}^{SR}$, and between $\tilde{h}_f$ and $h_f^{ATM}$, are shown in Table 3.

Comparison of $\tilde{h}_{fs}$ with $h_{fs}^{SR}$ tells us how well we can estimate snow depths, in the absence of a snow-radar, by differencing lidar ($h_f^{ATM}$) and radar freeboards ($h_{fi}^{CS2}$). As mentioned earlier, this is of particular interest since near

coincident observations of freeboards from CS-2 (a radar altimeter) and IS-2 (a lidar to be launched in late 2018) are potentially useful for providing large-scale estimates of snow depth (Kwok and Markus, 2017). The results indicate that the derived snow depth may be lower than expected, perhaps due to the processes discussed earlier.

When $h_f^{ATM}$ and the derived $\tilde{h}_{fs}$ are used to estimate ice thickness using Equation 7 (i.e., $h_i^{lidar}(h_f^{ATM}, \tilde{h}_{fs})$), the resulting ice cover is thicker because a larger fraction of the total freeboard is given to the higher density sea ice than to the lower density snow layer. For the three years, the derived transect-averaged ice thicknesses are (in meters): 2.77±1.06 and 2.78±1.10 (October, 2011), 3.17±0.97 (October 2014), and 2.54±0.90 (October 2016) (Table 2 and Figures 4&5). This is between 0.1 and 0.6 m thicker than those thicknesses discussed in Section 3: 2.40±1.07 and 2.51±1.16 (October, 2011), 2.60±1.15 m (October 2014), and 2.44±1.18 m (October 2016) (Table 2 and Figure 2).

## 5.4    Ice thickness assuming zero ice freeboard

Another approach to estimate ice thickness assumes that snow depth is equal to the total freeboard (Kurtz and Markus, 2012), i.e., zero ice freeboard. This is informed by the view that sea ice freeboards in field observations in the Antarctic usually have means very near zero. Here, we assess the potential effect of this assumption by comparing $h_i^{lidar}(h_f^{ATM}, h_f^{ATM})$ with $h_i^{lidar}(h_f^{ATM}, h_{fs}^{SR})$ (Equation 7). The differences for the three years show that the transect-averaged ice thicknesses could be potentially underestimated by up to a meter (Figure 6). So, in the interior of the Antarctic ice cover, perhaps it may be more useful to use estimates of snow depth when differences of freeboards are available (i.e., ICESat-2 and CS-2) than to assume zero ice freeboard.

## 6    Conclusions

In this paper, we addressed two topics using the Weddell Sea data set acquired by Operation IceBridge and CryoSat-2. First, we examined the spatial and interannual variability of total freeboard, snow depth, and sea ice thickness in three years (2011, 2014, and 2016) of repeated IceBridge surveys of a transect across the Weddell Sea. Second, the estimation of sea ice thickness, in the absence of snow depth measurements, using only freeboards from the IS-2 and near-coincident CS-2 radar freeboards is analyzed. Here, ATM lidar freeboard is used as a proxy of ICESat-2 freeboard. Presuming that CS-2 acquisitions are available after the launch of IS-2, the relevance of this analysis pertains to the use of differences between IS-2 and CS-2 freeboards as estimates of snow depth for ice sea thickness calculations.

Of geophysical interest are the following results:

- Averaged over this Weddell Sea transect, ice thickness ranges between 2.40±1.07 and 2.51±1.16  (October, 2011) and 2.60±1.15 m (October 2014), and 2.44±1.18 m (October 2016) (Table 2 and Figure 2). The average thicknesses are strikingly similar. As in the east-west gradient in snow depth and freeboard, the thicker sea ice is in the more deformed ice cover in the western Weddell Sea.  These OIB estimates are much higher than that reported by most drilling and ship-based measurements.

- Spatial/regional statistics of snow depth and ice thickness may be biased due to the incomplete sampling of the thin end of the snow depth distribution.

- For the three years, radar freeboards from CS-2 (i.e., uncompensated for snow thickness) sampled along the same transect are consistently higher (by up to 8 cm) than those computed using IceBridge data. This suggests radar scattering that originates above the snow-ice interface, likely associated with saline snow in the basal layers of the snow column reported in the literature (Section 5).

- When only differences in lidar (ICESat-2 or ATM) and radar (CS-2) freeboards are available for the calculation of sea ice thickness, the consequence of higher than expected radar freeboard is that the sea ice thicknesses are also higher (i.e., overestimated). This can be up to 0.6 m thicker than those thicknesses computed using snow depth from the snow radar.

- Results also show that using differences in lidar (ICESat-2 or ATM) and radar (CS-2) freeboards for the calculation of sea ice thickness is preferable to the approach that assumes that snow depth is equal to the total freeboard, where ice thickness here could be underestimated by up to a meter.

## Acknowledgments

We thank S. S. Pang for her software support during the course of this work. RK and SK carried out this work at the Jet Propulsion Laboratory, California Institute of Technology, under contract with the National Aeronautics and Space Administration.

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

**Table 1**

(a) Mean total freeboards ( $h_f^{ATM}$ , $\tilde{h}_f$ ) and ice freeboards ( $\tilde{h}_{fi}$ , $\tilde{h}'_{fi}$ ) along the Weddell Sea transect in 2011, 2014, and 2016 (Retrieved quantities from OIB and CS-2 only are in bold, the other quantities are derived)

| centimeters | $h_f^{ATM}$ | $\tilde{h}_f$ | $\tilde{h}_{fi}$ | $\tilde{h}'_{fi}$ |
|---|---|---|---|---|
| 2011 (Oct 12&25) | **49.3±17.2** | 56.3±20.3 | 19.9±10.1 | 18.7±8.89 |
| | **49.8±17.9** | 54.7±19.3 | 19.4±9.72 | 18.5±8.94 |
| 2014 (Oct 20) | **56.3±17.2** | 67.0±16.6 | 23.4±7.70 | 21.5±7.30 |
| 2016 (Oct 27) | **45.4±16.3** | 47.3±13.3 | 17.3±6.67 | 16.9±6.67 |

(b) Mean radar freeboard ( $h_{fi}^{CS2}$ , $\tilde{h}_{fi}^{CS2}$ ) and snow depth ( $h_{fs}^{SR}$ , $\tilde{h}_{fs}$ ) in 2011, 2014, and 2016

| centimeters | $h_{fi}^{CS2}$ | $\tilde{h}_{fi}^{CS2}$ | $h_{fs}^{SR}$ | $\tilde{h}_{fs}$ |
|---|---|---|---|---|
| 2011 (Oct 12&25) | **11.9±8.40** | 6.00±7.21 | **36.3±11.7** | 30.6±12.1 |
| | **11.7±8.10** | 7.66±7.73 | **35.2±11.0** | 31.1±12.1 |
| 2014 (Oct 20) | **13.8±6.28** | 5.39±6.48 | **43.6±10.2** | 34.9±12.0 |
| 2016 (Oct 27) | **10.7±5.91** | 9.52±8.89 | **30.0±8.51** | 28.3±12.1 |

**Table 2**

Mean sea ice thickness along the Weddell Sea loop from retrieved and derived freeboards and snow depths (2011, 2014, and 2016)

| meters | 2011 (Oct 12 and 25) | | 2014 (Oct 20) | | 2016 (Oct 27) | |
|---|---|---|---|---|---|---|
| | $h_{fs}^{SR}$ | $\tilde{h}_{fs}$ | $h_{fs}^{SR}$ | $\tilde{h}_{fs}$ | $h_{fs}^{SR}$ | $\tilde{h}_{fs}$ |
| $h_f^{ATM}$ | **2.40±1.07** | *2.77±1.06* | **2.60±1.15** | *3.17±0.97* | **2.44±1.18** | *2.54±0.90* |
| | **2.51±1.16** | *2.78±1.10* | | | | |
| $\tilde{h}_{fi}$ | 3.07±1.29 | 2.89±1.13 | 3.62±1.00 | 3.36±0.97 | 2.62±0.83 | 2.57±0.87 |
| | 2.99±1.23 | 2.87±1.14 | | | | |
| $\tilde{h}'_{fi}$ | *3.07±1.17* | *2.77±1.06* | *3.62±0.97* | *3.17±0.97* | *2.62±0.84* | *2.54±0.90* |
| | *2.99±1.16* | *2.78±1.10* | | | | |

Notes: 1. $h_i^{lidar}(h_f^{ATM},\tilde{h}_{fs}) = h_i^{radar}(\tilde{h}'_{fi},\tilde{h}_{fs})$ .

    2. Thickness estimates using only OIB retrievals are in bold.

    3. Thickness estimates using only OIB ATM and CS-2 freeboards (i.e., not using snow depths from the OIB snow radar) are in italics.

**Table 3**
Differences (mean and standard deviation) and correlation ($\rho$) between the derived and retrieved total freeboard, snow depth, radar (CS-2) freeboard, and ice freeboard in 2011, 2014, and 2016

| cm/$\rho$ | 2011 (Oct 12 and 25) | | 2014 (Oct 20) | | 2016 (Oct 27) | |
|---|---|---|---|---|---|---|
| $\tilde{h}_f$ vs $h_f^{ATM}$ | 7.00±11.7 | 0.81 | 10.6±9.79 | 0.83 | 1.93±9.32 | 0.82 |
| | 4.93±10.1 | 0.85 | | | | |
| $\tilde{h}_{fs}$ vs $h_{fs}^{SR}$ | -5.72±9.58 | 0.67 | -8.68±7.63 | 0.77 | -1.63±7.71 | 0.77 |
| | -4.16±8.67 | 0.72 | | | | |
| $h_{fi}^{CS2}$ vs $\tilde{h}_{fi}^{CS2}$ | 5.96±10.3 | 0.13 | 8.45±7.08 | 0.38 | 1.21±8.22 | 0.44 |
| | 4.05±9.31 | 0.30 | | | | |
| $\tilde{h}'_{fi}$ vs $\tilde{h}_{fi}$ | -1.26±2.10 | 0.98 | -1.91±1.67 | 0.97 | -0.35±1.69 | 0.96 |
| | -0.91±1.90 | 0.98 | | | | |

**Figure Captions**

Figure 1. Relationship between the different height quantities defined in the text.

Figure 2. Repeat surveys of the Weddell Sea Loop flight-line in 2011, 2014, and 2014. (a) Oct 11, 2011. (b) Oct 25, 2011. (c) Oct 20, 2014. (d) Oct 27, 2016. For each flight-track, we show the spatial distribution of ice thickness (top panel) and its along-track profile (middle panel), and the total freeboard and snow depth from the ATM lidar and snow radar on Operation IceBridge (bottom panel). Samples are 12.5 km averages.

Figure 3. Sampling of snow depth distributions and thickness estimates. (a) Oct 11, 2011. (b) Oct 20, 2014. (c) Oct 27, 2016. For each date, we show the relationships between snow depth and total freeboard (top left), the snow depth distributions (bottom left) and the thickness profiles (right) before (black) and after (red) filling the freeboard samples without snow depth retrievals. Dashed red line in top left panels show the presumed relationship between snow depth and total freeboard where snow depths are not resolved by the snow radar. Oct 25, 2011 is not shown here.

Figure 4. Comparisons of observed and derived snow depth, total freeboard, radar freeboard, ice thickness for (a) Oct 11, 2011. (b) Oct 25, 2011. Top panel: snow depth ($\tilde{h}_{fs}$ vs $h_{fs}^{SR}$). Second panel: total freeboard ($\tilde{h}_f'$ vs $h_f^{ATM}$). Third panel: ice freeboard ($\tilde{h}_{fi}^{CS2}$ vs $h_{fi}^{CS2}$). Bottom panel: ice thickness ($h_i^{radar}(\tilde{h}_{fi}', \tilde{h}_{fs})$ vs $h_i^{lidar}(h_f^{ATM}, h_{fs}^{SR})$). Note that $h_i^{lidar}(h_f^{ATM}, \tilde{h}_{fs}) = h_i^{radar}(\tilde{h}_{fi}', \tilde{h}_{fs})$. Snow depths from OIB ($h_{fs}^{SR}$) are not used in the derived estimates. Samples are 12.5 km averages.

Figure 5. Same as Figure 4 except for (a) Oct 20, 2014. (b) Oct 27, 2016.

Figure 6. Estimates of ice thickness assuming zero ice freeboard (i.e., $h_{fi} = 0$ or $h_i^{lidar}(h_f^{ATM}, h_{fs}^{SR})$ vs $h_i^{lidar}(h_f^{ATM}, h_f^{ATM})$ or $h_i^{radar}(0, h_f^{ATM})$). (a) Oct 11, 2011. (b) Oct 25, 2011. (c) Oct 20, 2014. (d) Oct 27, 2016. Samples are 12.5 km averages.

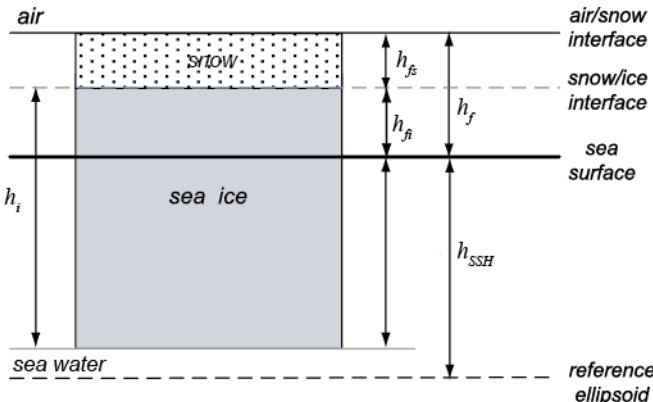

Figure 1.  Relationship between the different height quantities defined in the text.

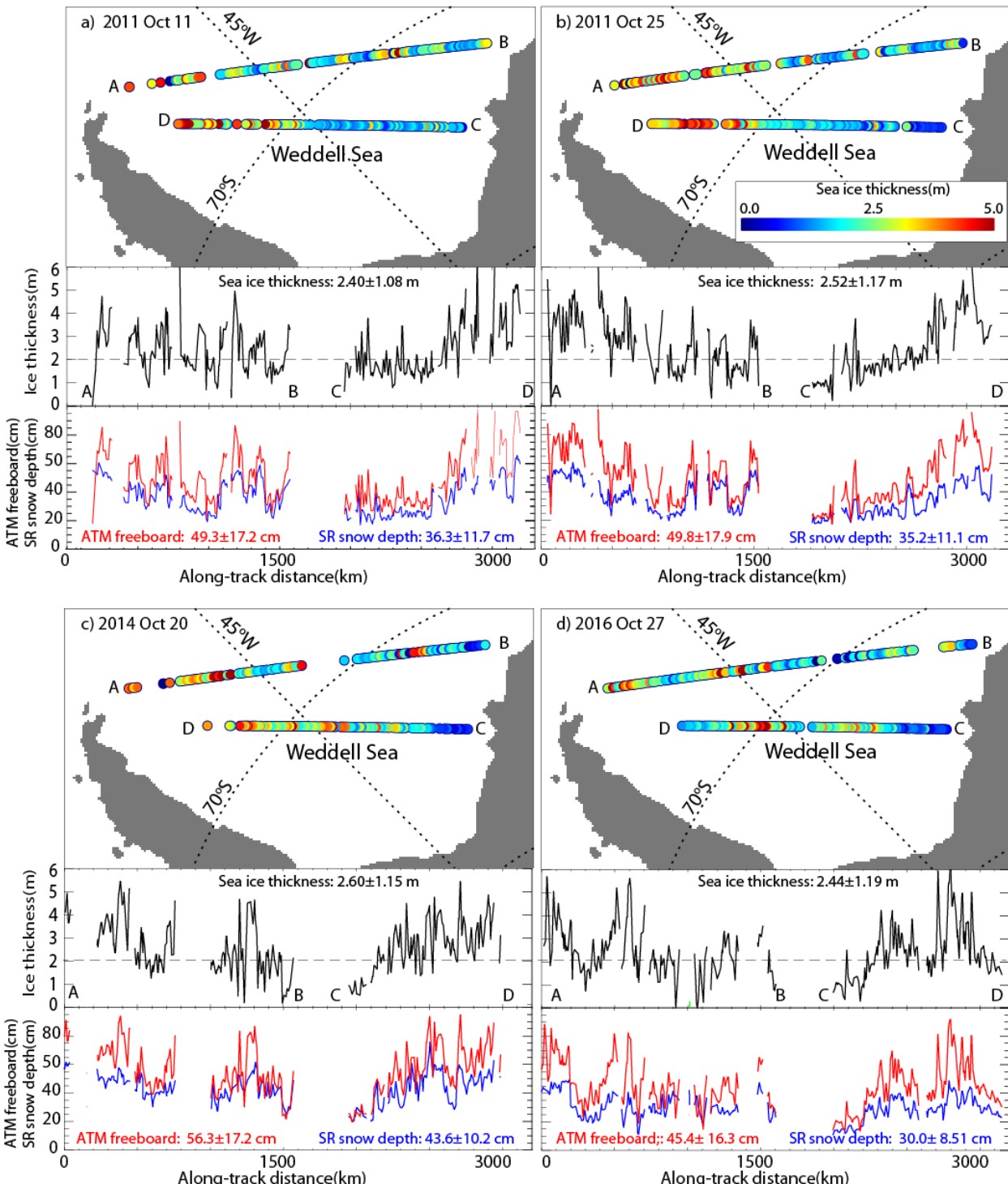

Figure 2. Repeat surveys of the Weddell Sea Loop flight-line in 2011, 2014, and 2014. (a) Oct 11, 2011. (b) Oct 25, 2011. (c) Oct 20, 2014. (d) Oct 27, 2016. For each flight-track, we show the spatial distribution of ice thickness (top panel) and its along-track profile (middle panel), and the total freeboard and snow depth from the ATM lidar and snow radar on Operation IceBridge (bottom panel). Samples are 12.5 km averages.

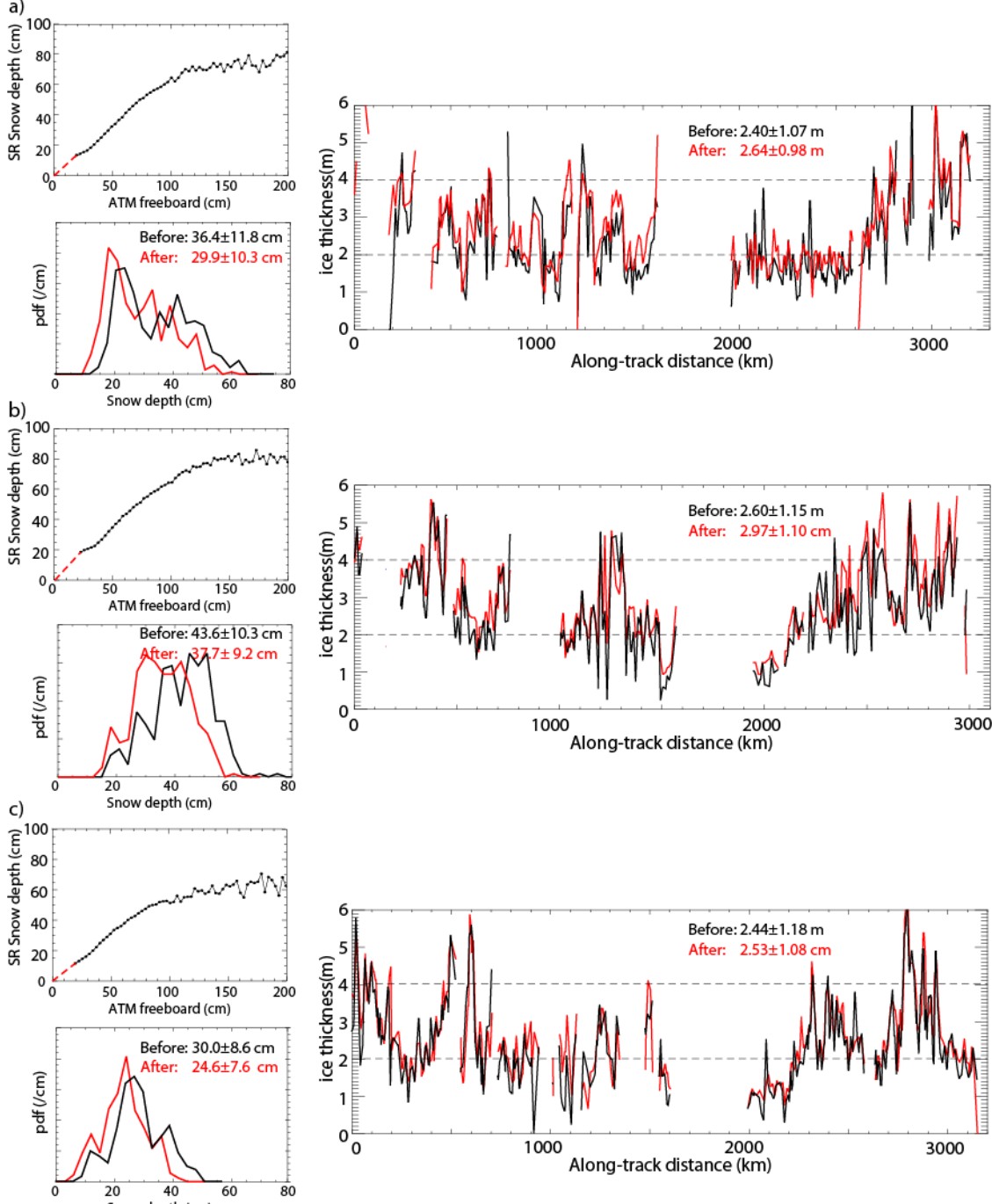

Figure 3. Sampling of snow depth distributions and thickness estimates. (a) Oct 11, 2011. (b) Oct 20, 2014. (c) Oct 27, 2016. For each date, we show the relationships between snow depth and total freeboard (top left), the snow depth distributions (bottom left) and the thickness profiles (right) before (black) and after (red) filling the freeboard samples without snow depth retrievals. Dashed red line in top left panels show the presumed relationship between snow depth and total freeboard where snow depths are not resolved by the snow radar. Oct 25, 2011 is not shown here.

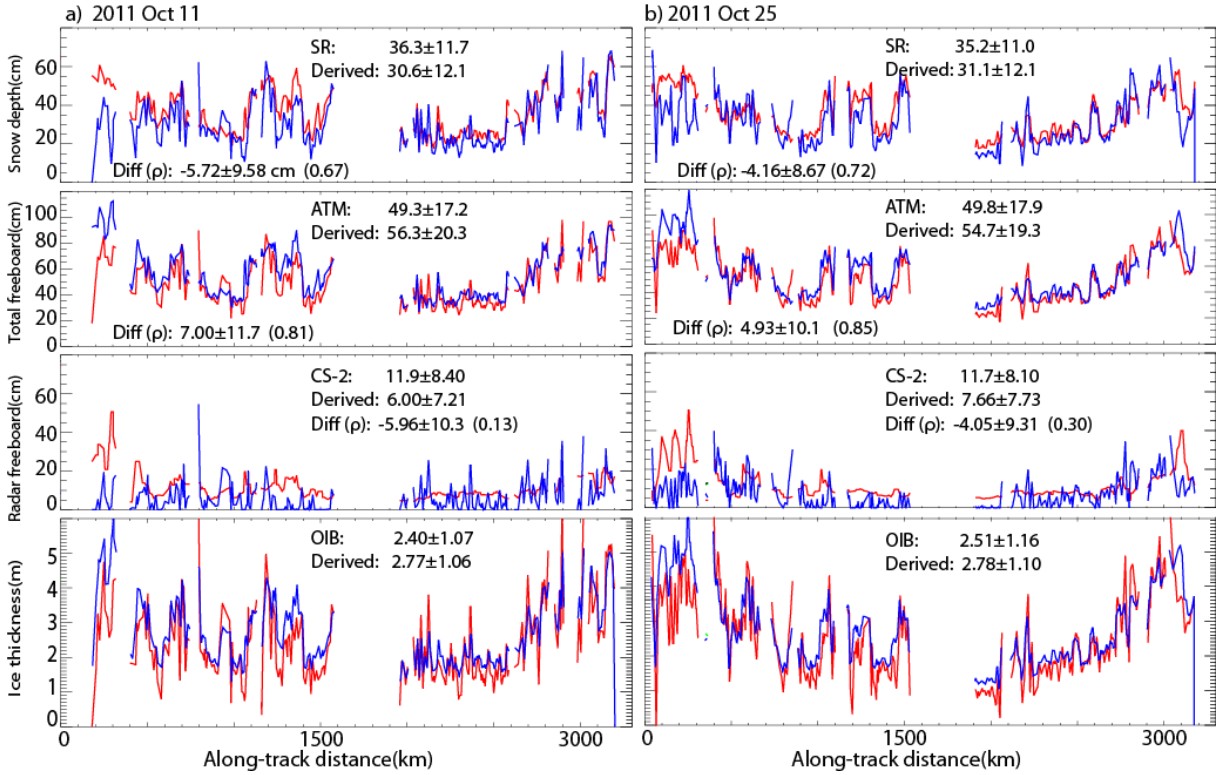

Figure 4. Comparisons of observed and derived snow depth, total freeboard, radar freeboard, ice thickness for (a) Oct 11, 2011. (b) Oct 25, 2011. Top panel: snow depth ($\tilde{h}_{fs}$ vs $h_{fs}^{SR}$). Second panel: total freeboard ($\tilde{h}'_{f}$ vs $h_{f}^{ATM}$). Third panel: ice freeboard ($\tilde{h}_{fi}^{CS2}$ vs $h_{fi}^{CS2}$). Bottom panel: ice thickness ($h_{i}^{radar}(\tilde{h}'_{fi},\tilde{h}_{fs})$ vs $h_{i}^{lidar}(h_{f}^{ATM},h_{fs}^{SR})$). Note that $h_{i}^{lidar}(h_{f}^{ATM},\tilde{h}_{fs})=h_{i}^{radar}(\tilde{h}'_{fi},\tilde{h}_{fs})$. Snow depths from OIB ($h_{fs}^{SR}$) are not used in the derived estimates. Samples are 12.5 km averages.

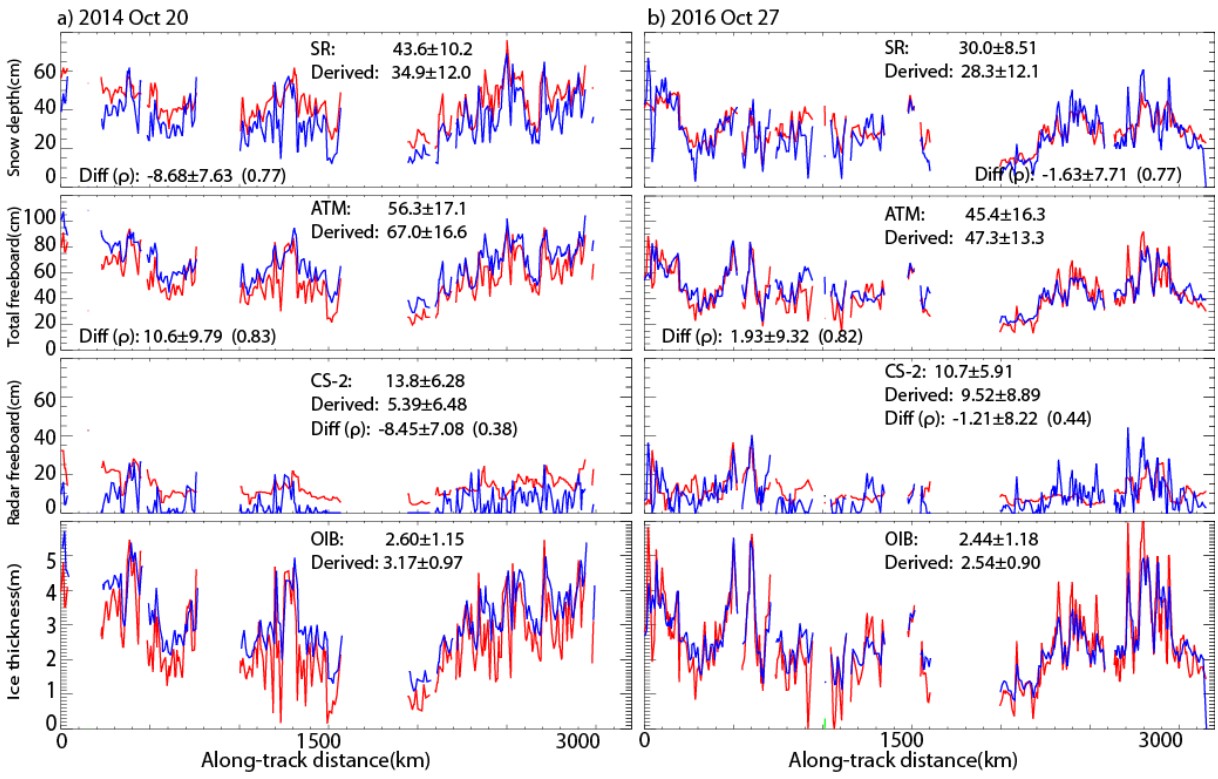

Figure 5. Same as Figure 3 except for (a) Oct 20, 2014. (b) Oct 27, 2016

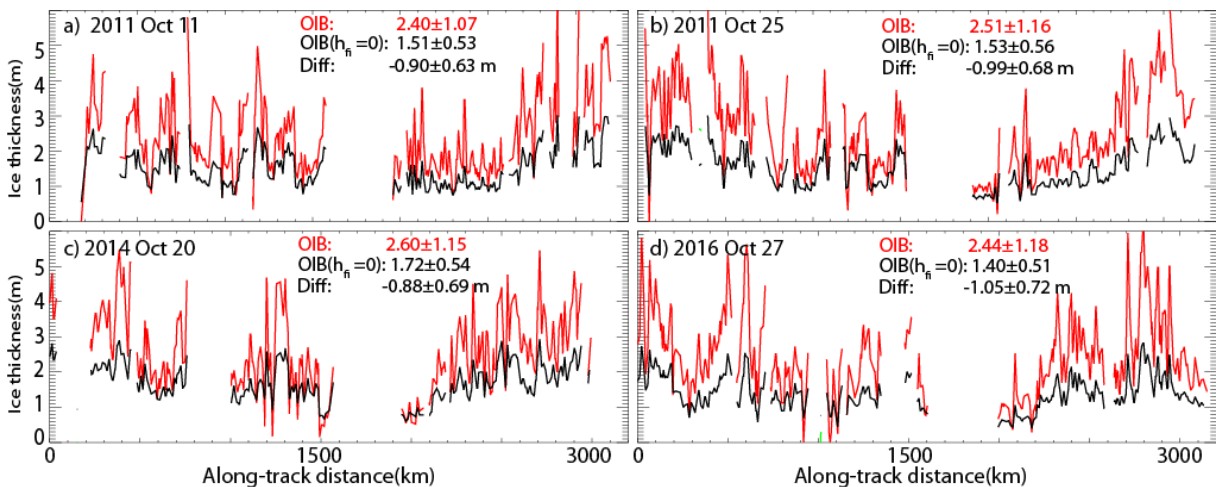

Figure 6. Estimates of ice thickness assuming zero ice freeboard (i.e., $h_{fi} = 0$ or $h_i^{lidar}(h_f^{ATM}, h_{fs}^{SR})$ vs $h_i^{lidar}(h_f^{ATM}, h_f^{ATM})$ or $h_i^{radar}(0, h_f^{ATM})$). (a) Oct 11, 2011. (b) Oct 25, 2011. (c) Oct 20, 2014. (d) Oct 27, 2016. Samples are 12.5 km averages.