# Peer review of "Three years of sea ice freeboard, snow depth, and ice thickness of the Weddell Sea from Operation IceBridge and CryoSat-2"

_The Cryosphere, 2018_

## Referee Comment (RC1) · Anonymous Referee #1 · 3 Jul 2018

The paper is almost flawless, with results and discussions complete. It is also very well organized and structured making the case study approach easy to follow and make comparisons. However, there are few confusing sections which could be clarified (see below).

1) Introduction

(Page 1, lines 1 to 8): One of the biggest differences between the Arctic and the Antarctic sea ice is with its associated snow cover (eg. how thick the snow cover is and its corresponding variable geophysical properties including its freeboard). I think the authors should briefly summarize the observed differences in snow covers on Antarctic

versus Arctic sea ice and how they likely influence (and complicate) ice freeboard and thickness retrievals from radar altimetry.

Line 13: "Remote sensing issues" ?? The author needs to briefly describe what these issues are. "to determine freeboard" snow or ice?

Page 3: "Data description" should be 2, not 1. And section numbering should be corrected throughout thereafter.

Page 4 and throughout thereafter: The authors should really consider simplifying the symbol notations if possible. It looks very confusing, small and complicated. Too many subscripts within a subscript (for e.g. snow depth from the snow radar in line 25). This causes sections 2.2 and 2.3 to be read in a very unclear and confusing manner.

Table 3: Even though the table shows differences and correlation between derived and measured freeboard, the "/" symbol is confusing. Please consider changing.

Figure 2: No coordinate info in the map figures for the flight lines. Something needs to be added for visual geo-referencing.

---

## Referee Comment (RC2) · Anonymous Referee #2 · 12 Jul 2018

The paper presents the variability of sea ice and snow parameters on two repeated OIB survey lines across the Weddell Sea and examines the potential synergism between OIB lidar and CS-2 radar. The paper is well written and can be published with some minor revisions.

The authors use data from three sensors: CS-2 radar, OIB LIDAR ATM and Snow Radar (SR), which have vastly different spatial resolutions and data collection date/time. In the "Data Description" section, the authors provide some background information about each sensor/dataset used but don't include sufficient details on how these data sets are matched up and their associated mismatching uncertainties (spatial

and temporal). In addition, the authors made many comparisons between the "derived estimates" and "retrieved quantities." What is missing from those comparisons are consistencies between CS-2 and SR ice freeboard, as well as ATM and SR snow (total) freeboard, which is fundamental to the differences between the "derived estimates" and "retrieved quantities."

Page 3, Ln 32: It is not clear "what "aspect of the algorithm" has been disabled.

Page 4, Ln 1-11: A detailed and quantitative description of the interface detection algorithm is necessary but missing. Also, please provide references if available.

Page 4, Ln 24: "described above" should be "described below."

Page 6, Ln 14-15: The sea ice thickness are calculated in six different ways. Can the authors compare this calculated thickness against the ice thickness derived from the snow radar data only?

Page 7, Ln 22-23: It's fine to compare total freeboard against snow depth, but comparisons of ATM and SR freeboards should be included in the discussion.

Page 9, Ln 10-11: When comparing monthly CS-2 data against individual OIB track data, one needs to understand the variability of sea ice at month scale. This discussion should be included in the paper.

―――――――――――――――――

---

## Author Comment (AC1) · 27 Jul 2018

**Anonymous Referee #1 (Referee comments are in italics)**

*The paper is almost flawless, with results and discussions complete. It is also very well organized and structured making the case study approach easy to follow and make comparisons. However, there are few confusing sections which could be clarified (see below).*

We thank the reviewer for his or her time in reviewing the manuscript and providing helpful feedback. The suggestions have significantly clarified the text and figures; we are appreciative of your help in improving the manuscript.

*1) Introduction*
*(Page 1, lines 1 to 8): One of the biggest differences between the Arctic and the Antarctic sea ice is with its associated snow cover (eg. how thick the snow cover is and its corresponding variable geophysical properties including its freeboard). I think the authors should briefly summarize the observed differences in snow covers on Antarctic versus Arctic sea ice and how they likely influence (and complicate) ice freeboard and thickness retrievals from radar altimetry.*

We have added a few sentences to the first paragraph (below) to highlight the differences between Arctic and Antarctic snow depths and ice thickness. But, we point the reviewer/reader to perhaps one of the best summaries of Antarctic snow depth (Massom et al., 2001).

Added text: "…One distinguishing feature of sea ice in the Antarctic vis-à-vis the Arctic is the common occurrence of snow-ice due to heavier snow fall (Massom et al., 2001): when the snow load depresses the ice surface of the thinner Antarctic sea ice below sea level, seawater infiltrating the base of the snow layer leads to the formation snow-ice when the resultant slush freezes. The thicker snow cover, flooding, and snow-to-ice conversion (in addition basal freezing) complicate the large-scale retrievals of snow depth and ice thickness…."

*Line 13: "Remote sensing issues" ?? The author needs to briefly describe what these issues are. "to determine freeboard" snow or ice?*

The follow text has been added to clarify the meaning:

"…of ICESat and the launch of the ICESat-2 (IS-2) lidar this year, OIB has acquired a unique time series of that allows for examination of the interannual behavior of Antarctic sea ice cover as well as a better understanding of the remote sensing issues associated with the retrieval of sea ice freeboard and thickness. In addition to a lidar to determine freeboard, the OIB instrument suite includes an ultra-wideband radar that is capable of resolving the location of the air-snow and snow-ice interfaces, and hence providing snow depth estimates. Over the OIB mission, the sensitivity of snow depth retrievals to

associated snow properties (density and salinity) has become a special emphasis because of the significant impact of snow on thickness estimates using lidar or radar (Kwok, 2014)."

*Page 3: "Data description" should be 2, not 1. And section numbering should be corrected throughout thereafter.*

Corrected.

*Page 4 and throughout thereafter: The authors should really consider simplifying the symbol notations if possible. It looks very confusing, small and complicated. Too many subscripts within a subscript (for e.g. snow depth from the snow radar in line 25). This causes sections 2.2 and 2.3 to be read in a very unclear and confusing manner.*

Yes, we recognize that it is somewhat confusing but we think that they are needed for the discussion in the text. In order to make it more accessible to the reader, we've done the following:

1. At the beginning of Section 3, the differences between retrieved and derived quantities are defined. Added text: "…Here, we define derived versus retrieved quantities. Derived estimates, referred to in the balance of this paper and this section, are quantities calculated from the retrieved quantities. The significance of these derived estimates is discussed…"

2. Wherever possible in the text, we use the full text description of a variable in addition to its symbolic notation. For example, instead of,

"…Comparison of $\tilde{h}_{fs}$ with $h_{fs}^{SR}$ tells us how well we can estimate snow depths using the differences between $h_f^{ATM}$ and $h_{fi}^{CS2}$ in the absence of a snow-radar…."

We substitute with:

"…Comparison of the derived ($\tilde{h}_{fs}$) with the retrieved snow depth ($h_{fs}^{SR}$) tells us how well we can estimate snow depths using the differences between lidar ($h_f^{ATM}$) and radar freeboards ($h_{fi}^{CS2}$) in the absence of a snow-radar…"

*Table 3: Even though the table shows differences and correlation between derived and measured freeboard, the "/" symbol is confusing. Please consider changing.*

A separate column added, for the correlation values, has been added to Table 3 so that a "/" is no longer needed.

*Figure 2: No coordinate info in the map figures for the flight lines. Something needs to be added for visual geo-referencing.*

Latitudes and longitude labels have been added to Figure 2.

---

## Author Comment (AC2) · 27 Jul 2018

**Responses to Anonymous Referee #2 (Referee comments are in italics)**

*The paper presents the variability of sea ice and snow parameters on two repeated OIB survey lines across the Weddell Sea and examines the potential synergism between OIB lidar and CS-2 radar. The paper is well written and can be published with some minor revisions.*

We thank the reviewer for his or her time in reviewing the manuscript and providing helpful feedback. The suggestions have significantly clarified the text and figures; we are appreciative of your help in improving the manuscript.

*The authors use data from three sensors: CS-2 radar, OIB LIDAR ATM and Snow Radar (SR), which have vastly different spatial resolutions and data collection date/time. In the "Data Description" section, the authors provide some background information about each sensor/dataset used but don't include sufficient details on how these data sets are matched up and their associated mismatching uncertainties (spatial and temporal). In addition, the authors made many comparisons between the "derived estimates" and "retrieved quantities." What is missing from those comparisons are consistencies between CS-2 and SR ice freeboard, as well as ATM and SR snow (total) freeboard, which is fundamental to the differences between the "derived estimates" and "retrieved quantities."*

In this paper, we start with three independent retrieved quantities: ATM total freeboard, snow depth from the snow radar, and radar freeboard from the CS-2. The aim was to attempt to understand the differences between the retrieved quantities and the derived quantities, and their relative impact on ice thickness estimates.

Specifically, we compared the use of:
1. retrieved snow depth and derived snow depth.
2. retrieved total freeboard and derived total freeboard.
3. retrieved radar freeboard and calculated radar freeboard.

The relative consistency of both the retrieved and derived quantities can be found in Table 1 and Table 3. Their impact on thickness estimates can be found in Table 2. Our intent is clarified in the text. We have added text to clarify the overall intention in this regard.

We also note that snow radar freeboard is not used in this work. There has been less attention paid to snow radar freeboard to date and is not addressed here.

*Page 3, Ln 32: It is not clear "what "aspect of the algorithm" has been disabled.*

Sentence has been modified to read: "…the aspect of the algorithm that deals with system sidelobes has been disabled.."

*Page 4, Ln 1-11: A detailed and quantitative description of the interface detection algorithm is necessary but missing. Also, please provide references if available.*

The details of the interface detection algorithm is described in Kwok and Maksym {2014) and briefly summarized in the text.

*Page 4, Ln 24: "described above" should be "described below."*

To be clear, the phrase has been revised to read: "… described in Section 2…"

*Page 6, Ln 14-15: The sea ice thicknesses are calculated in six different ways. Can the authors compare this calculated thickness against the ice thickness derived from the snow radar data only?*

Yes, it is theoretically possible but locating the sea surface in the snow radar has not been attempted and beyond the current scope of the manuscript.

*Page 7, Ln 22-23: It's fine to compare total freeboard against snow depth, but comparisons of ATM and SR freeboards should be included in the discussion.*

The ATM lidar heights, when referenced to the local sea surface, are the total freeboard. And, snow depth is an independent estimate from Operation IceBridge. This is clarified in the text. We added a sentence at the beginning of the section as a reminder to the reader: "…As a reminder, snow depth is the retrieved parameter from the snow radar ($h_{fs}^{SR}$) and total freeboard is from the ATM lidar ($h_{f}^{ATM}$)…"

*Page 9, Ln 10-11: When comparing monthly CS-2 data against individual OIB track data, one needs to understand the variability of sea ice at month scale. This discussion should be included in the paper.*

The monthly variability of the CS-2 radar freeboards can found in Table 1 in the original manuscript. As suggested, we have included this variability in of the observed and the calculated radar freeboards in the text of the paper.

---

## Author Response (AR2)

The editor's corrections were addressed, and we corrected several typographical errors.